# Humanized Monoclonal Antibody Against Citrullinated Histone H3 Attenuates Myocardial Injury and Prevents Heart Failure in Rodent Models

**DOI:** 10.3390/biom15081196

**Published:** 2025-08-20

**Authors:** Matthew Weber, Yuchen Chen, Xinyu Zhou, Heejae Chun, Di Wu, Ki Ho Park, Chuanxi Cai, Yongqing Li, Jianjie Ma, Zequan Yang

**Affiliations:** 1Department of Surgery, University of Virginia, Charlottesville, VA 22903, USA; zds7kb@uvahealth.org (M.W.); dvj9dr@virginia.edu (Y.C.); ywx5wp@virginia.edu (X.Z.); fzt6dh@virginia.edu (H.C.); dw6a@virginia.edu (D.W.); uuh2qa@virginia.edu (K.H.P.); txt5pt@virginia.edu (C.C.); 2Department of Surgery, University of Michigan Health System, Ann Arbor, MI 48109, USA; yqli@med.umich.edu

**Keywords:** cardio-protection, immune modulation, NETosis, therapeutic development

## Abstract

**Background:** Excessive formation of neutrophil extracellular traps (NETs) leads to NETosis, accompanied by the release of citrullinated histone H3 (CitH3), a key mediator of septic inflammation. However, the role of CitH3 in sterile inflammation, such as acute myocardial infarction (MI) and post-MI heart failure, remains incompletely understood. **Methods and Results:** We investigated the role of CitH3, a byproduct of NETosis, in myocardial ischemia/reperfusion (I/R) injury using a murine MI model. C57BL/6J mice were subjected to left coronary artery (LCA) occlusion followed by reperfusion and treated with either a humanized anti-CitH3 monoclonal antibody (hCitH3-mAb) or control human IgG. In mice undergoing 40 min of LCA occlusion and 24 h of reperfusion, hCitH3-mAb administered 10 min before reperfusion significantly reduced infarct size by 36% compared to control (*p* < 0.05). Plasma levels of CitH3, IL-1β, and interferon-β were significantly elevated following MI but were attenuated by hCitH3-mAb. In addition, plasma and cardiac tissue from treated mice showed significantly lower levels of citrate synthase, a marker of mitochondrial injury, suggesting that hCitH3-mAb preserved mitochondrial integrity after MI. In mice undergoing 50 min of LCA occlusion and 21 days of reperfusion, longitudinal echocardiography revealed preservation of left ventricular ejection fraction (LVEF) in hCitH3-mAb-treated mice, with significant improvement observed on days 7, 14, and 21 post-MI (*p* < 0.05 vs. control). hCitH3-mAb also mitigated myocardial fibrosis and preserved tissue architecture. **Conclusions:** These findings demonstrated CitH3 as a critical mediator of myocardial injury and adverse remodeling following acute MI. Neutralization of CitH3 via hCitH3-mAb attenuates I/R injury and preserves cardiac function by mitigating inflammation and protecting mitochondrial integrity. Targeting CitH3 represents a promising therapeutic strategy to prevent heart failure following MI.

## 1. Introduction

Ischemic heart disease remains the leading cause of death in the United States, responsible for approximately one in five deaths [1]. Among its clinical manifestations, myocardial infarction (MI) and heart failure (HF) account for the majority of disease-related morbidity and mortality. Inflammatory responses play a crucial role in both myocardial necrosis during early ischemia/reperfusion injury (IRI) and in post-MI left ventricular (LV) remodeling, which often progresses to HF. Neutrophils, the body’s earliest responders in both septic and sterile inflammatory conditions [2,3,4,5], are recruited to the infarcted myocardium within hours, guided by cellular debris and damage-associated molecular patterns (DAMPs) generated by necrotic cells [6,7]. Even minor variations in the duration or extent of neutrophil infiltration can markedly influence cardiac architecture, with the potential for severe downstream consequences [8]. Numerous experimental strategies targeting neutrophil recruitment or activation have demonstrated protective effects in preclinical models, yet clinical translation has thus far been unsuccessful [5,9,10]. This reflects a combination of factors, including an incomplete understanding of the mechanisms involved, the influence of coexisting risk factors, and the difficulty of accurately defining the optimal therapeutic window [5,9,10].

Activated neutrophils formed neutrophil extracellular traps (NETs), which help to capture the pathogens for macrophages to clear the debris at the injured area and allow tissue repair processes to begin. However, excessive formation of NETs leads to NETosis, which can damage host tissue by releasing cytotoxic components like citrullinated histones (CitH3) [10,11,12,13]. Persistent NETosis can sustain and amplify inflammatory responses, contributing to tissue damage and fibrosis in diseases like heart failure [14,15]. Emerging studies underscore the significance of CitH3 in mediating NETs-driven immune activation and subsequent tissue injury [16,17,18,19,20,21,22,23,24,25,26]. We have found that CitH3, released during NETosis, contributed to the inflammatory responses during septic shock, and neutralization of CitH3 attenuated inflammatory responses and improved survival of septic animals [16,27]. Furthermore, we have developed a novel mouse CitH3 monoclonal antibody (CitH3-mAb) with superior binding affinity and capacity compared to commercially available CitH3 antibodies [28]. We have recently successfully humanized the CitH3-mAb (hCitH3-mAb), making it suitable for large-scale production and potential human applications [29,30]. However, the role of NETosis, as well as CitH3, in sterile inflammation, such as acute MI and post-MI heart failure, is largely unknown.

In the current study, we found that plasma CitH3 level elevated during both early post-ischemia–reperfusion (within 24 h) and late post-infarct LV remodeling. Administration of hCitH3-mAb at the onset of reperfusion reduced myocardial infarct size, and repetitive dosing of hCitH3-mAb after reperfusion improved LVEF and LV remodeling.

## 2. Materials and Methods

This study complied with the 2011 Guide for the Care and Use of Laboratory Animals, 8th edition, as recommended by the U.S. National Institutes of Health, ensuring that all animals received humane care. The University of Virginia Animal Care and Use Committee reviewed and approved the study protocol (Protocol number: 4410-10-22). All efforts were made to minimize animal suffering and reduce the number of animals used in the experiments.

### 2.1. hCitH3-mAb Generation

To create the humanized hCitH3-mAb from the original mouse antibody [28], a multi-step approach was used to maintain binding specificity while reducing immunogenicity for human use. The detailed procedure for the scaled-up production of hCitH3-mAb has recently been accepted for publication in Nature Communications [30]. Briefly, hCitH3-mAb with >99.5% purity was purified from CHO cells and used for the current study.

### 2.2. Acute Myocardial Ischemia/Reperfusion Injury

Fourteen C57BL/6 wild-type male mice (8–12 weeks, purchased from The Jackson Laboratory, Bar Harbor, ME, USA) were used in the study. All mice underwent 40 min of left coronary artery (LCA) occlusion (ischemia) followed by 24 h of reperfusion. Mice were treated at 10 min before reperfusion either with an IV bolus of human IgG (5 mg/kg, control group, *n* = 6) or hCitH3-mAb (5 mg/kg, treated group, *n* = 8) via external jugular vein. Blood was obtained at 24 h of reperfusion by puncturing the right ventricle. Myocardial infarct size was evaluated at the end of 24 h of reperfusion by TTC-blue staining.

### 2.3. Post-Infarct Heart Failure Study

A total of 52 C57BL/6 wild-type mice (male, 8–12 weeks) underwent 50 min of LCA occlusion. 10 min after reperfusion, Mice were treated i.v. bolus with hCitH3-mAb (5 mg/kg, Treated group, *n* = 10) or human IgG (5 mg/kg, Control group, *n* = 10) via the external jugular vein. Then these mice underwent redosing of hCitH3-mAb or human IgG on post-infarct day 3, 7, and 15. 10 mice in each group underwent echocardiography 2 days before MI and on post-infarct day 1, day 7, day 14, and day 21, with blood and cardiac tissue harvested at day 21. Paralleling groups of mice were similarly treated and euthanized on days 1, 3, 7, and 14 post-MI to obtain blood and heart samples for biochemical and histological assessments (4 mice per group and per timepoint, IgG control and hCitH3-mAb treated).

### 2.4. Surgical Procedures to Induce Myocardial Infarction in Mice

Surgical procedure was performed as we previously reported [31]. Briefly, mice were anesthetized using 3% isoflurane and maintained at 1.5–2% isoflurane to keep heart and respiration rates consistent and placed in a supine position. They were orally intubated with a piece of PE-60 tube. Respiration was maintained with a rodent ventilator with room air, a frequency of 130 strokes/min, and a tidal volume of 6–8 μL/g weight. The hair of the chest and neck was removed using clippers and sterilely prepped and draped. The left pleural cavity was approached by cutting the left 3rd and 4th ribs and intercostal muscles with scissors, thus exposing the heart. An 8-0 Prolene suture was passed underneath the LCA at 1 mm inferior to the left atrium, then tied down over a short piece of PE-60 tubing to occlude the LCA for 40 or 50 min. Significant ECG changes, including widening of QRS and elevation of ST segment complex (monitored with a PowerLab data recording unit developed byADInstruments, Colorado Springs, CO, USA), together with color changes in the risk region were used to confirm successful LCA occlusion. Reperfusion was achieved by untying the ligature and removing the PE-60 tubing. Ketoprofen will be given subcutaneously at a dose of 4 mg/kg after reperfusion to alleviate post-operative pain. Core body temperature was monitored throughout the operation with a rectal thermocouple interfaced to a digital thermometer (Barnant Co., Barrington, IL, USA) and was maintained between 36.0 and 37.0 °C with a heating lamp.

### 2.5. Determination of Myocardial Infarct Size

At the end of 24 h of reperfusion, mice were euthanized under deep anesthesia, and the heart was excised, cannulated through the ascending aorta with a blunted 23-gauge needle, and perfused with 3 mL 37 °C 1% TTC in PBS (pH = 7.4). The LCA was then re-occluded by re-tying the suture left around the LCA. The heart was then perfused with 0.3–0.5 mL 10% Phthalo Blue (Heubach Ltd., Fairless Hills, PA, USA) to delineate the non-ischemic tissue. The left ventricle was cut into 5–7 transverse slices and fixed in 10% neutral buffered formalin solution. Each slice was weighed and photographed. The sizes of the non-ischemic area, the risk region, and the infarct area were calculated as a percentage of the corresponding slice multiplied by the weight of the slice [31,32].

### 2.6. Echocardiography (ECHO) Image Acquisition and Longitudinal Evaluation of Cardiac Function

Transthoracic echocardiography was conducted on mice using the Vevo F2 LT preclinical imaging system (Fujifilm Visualsonics, Toronto, ON, Canada) equipped with a UHF-46x probe. Anesthesia was induced with 3% isoflurane and maintained at 1.5–2% to ensure stable cardiac and respiratory rates across animals. Throughout the procedure, body temperature was regulated using a heated platform. For each mouse, three imaging views were obtained: B-mode parasternal long axis (LAX), B-mode short axis (SAX), and M-mode SAX, the latter captured at the level of the papillary muscles. Acquired images were analyzed in Vevo LAB. For M-mode SAX, measurements were taken from three sequential cardiac cycles occurring between breaths.

### 2.7. Western Blot

Plasma was separated from whole blood by centrifugation. Myocardial tissues were lysed in RIPA buffer (Thermo Fisher Scientific, Waltham, MA, USA) with protease and phosphatase inhibitors, and citrate synthase activity was assessed in different myocardial regions. Protein concentrations were determined by Pierce^TM^ BCA assay (Thermo Fisher Scientific, Waltham, MA, USA). Equal protein (20 µg) or plasma (3 µL) samples were resolved on 10% SDS–PAGE gels, transferred to PVDF membranes (Millipore, Burlington, MA, USA), and blocked in 5% BSA/TBST for 1 h at room temperature. Membranes were incubated overnight at 4 °C with primary antibodies against IL-1β (Abcam, Waltham, MA, USA), IFN-β (Abcam), citrate synthase (Abcam), or CitH3 (hCitH3-mAb), followed by HRP-conjugated secondary antibodies for 1 h. Signals were detected using ECL reagents (Millipore) and visualized with a chemiluminescence imager (Amersham Imager 680, GE Life Sciences, Marlborough, MA, USA). Densitometric analysis was performed using ImageJ (Version 1.54f, NIH, Bethesda, MD, USA).

### 2.8. Collagen Deposition in Post-Mi Myocardium

Masson’s Trichrome staining was performed to assess tissue fibrosis. Formalin-fixed, paraffin-embedded heart tissues were sectioned at 4–5 µm thickness and mounted on glass slides. After deparaffinization and rehydration through graded alcohols to distilled water, slides were stained using a standard Masson’s Trichrome staining protocol (Sigma-Aldrich HT15 kit, Sigma-Aldrich, St. Louis, MO, USA). Fibrosis quantification was performed by analyzing the blue collagen staining to determine the extent of fibrosis relative to the overall tissue area using ImageJ software (Version 1.54f).

### 2.9. Statistical Analysis

Group comparisons were analyzed using one-way ANOVA with Bonferroni post hoc correction or unpaired Student’s *t*-test; paired Student’s *t*-test was used to analyze changes in heart rate. Statistical analyses were performed in Prism (Version 7.05, GraphPad Software, La Jolla, CA, USA). Data were presented as Mean ± SEM in Figure 1 and Figure 2, as Mean ± SD in Figure 3 and Figure 4, and *p* < 0.05 was considered significant.

## 3. Results

### 3.1. hCitH3-mAb Attenuates Acute Myocardial IRI Injury

Recent studies highlight the critical role of CitH3 in NETosis-induced immune responses and tissue injury. However, the role of CitH3 in sterile inflammation during MI remains poorly understood. We hypothesize that CitH3 contributes to myocardial ischemia–reperfusion injury (IRI) through NETosis and associated inflammatory cascades, and that targeting CitH3 with a neutralizing monoclonal antibody (hCitH3-mAb) could limit infarct expansion and prevent the progression to HF.

C57BL/6J mice underwent left coronary artery (LCA) occlusion (40 min ischemia) followed by 24 h of reperfusion and received either hCitH3-mAb or human IgG control intravenously 10 min prior to reperfusion. Risk region analysis confirmed similar areas at risk between groups, validating equivalent ischemic insult. However, hCitH3-mAb treatment significantly reduced infarct size by ~36% compared to IgG controls (39 ± 4% vs. 62 ± 3%, Figure 1A,B). This protection was associated with reduced systemic inflammation, as evidenced by significantly lower serum IL-1β and IFN-β levels at 24 h post-reperfusion (Figure 1C,D), suggesting mitigation of acute NET-mediated immune activation.

### 3.2. Mitochondrial Protection by hCitH3-mAb

In the context of MI, mitochondrial integrity is critically compromised by mitochondrial damage. Citrate synthase (CS), a central enzyme in the citric acid cycle, serves as a marker of mitochondrial abundance and functionality, and indicates the degree of mitochondrial injury [33]. Following 40 min of LCA occlusion, myocardial CS levels were significantly reduced in the infarct and marginal zones by day 3 post-MI. hCitH3-mAb treatment preserved CS levels in these zones (Figure 2A). Correspondingly, plasma CS levels were elevated in control mice at days 1, 3, and 7 post-MI but were significantly lower in hCitH3-mAb-treated mice (Figure 2B), indicating protection of mitochondrial integrity.

### 3.3. hCitH3-mAb Preserves Post-MI Cardiac Function

Long-term effects were examined in C57BL/6J mice subjected to 50 min of LCA occlusion and 21 days of reperfusion. The treatment group received hCitH3-mAb (5 mg/kg, IV) at reperfusion onset and on days 3 and 7, whereas controls were given IgG. Serial echocardiographic assessments were carried out before surgery and on days 1, 7, 14, and 21 post-MI.

Serum CitH3 levels peaked immediately after infarction and declined over time, but were significantly lower in treated mice by day 14 (*p* < 0.01, Figure 3A). Treated mice demonstrated recovery of ejection fraction (EF) after day 1 nadir, while EF continued to decline in control animals. EF was significantly higher in the hCitH3-mAb group on day 7 (41.9 ± 5.1% vs. 25.9 ± 5.9%), day 14 (44.6 ± 6.2% vs. 25.6 ± 5.1%), and day 21 (46.2 ± 5.5% vs. 23.3 ± 4.7%, *p* < 0.01, Figure 3B). Both end-systolic and end-diastolic LV volumes were elevated and continued to trend throughout the 4-week period of observation in control and treated groups. The increase in LV volumes was higher in Control than hCitH3-mAb-treated mice (*p* = NS, Figure 3C).

### 3.4. hCitH3-mAb Mitigates Adverse Cardiac Remodeling

At 21 days post-reperfusion, hearts were harvested and stained with Masson’s trichrome to evaluate fibrosis. Mice treated with hCitH3-mAb exhibited significantly less ventricular dilation and myocardial fibrosis compared to controls (2.7 ± 0.8% vs. 5.7 ± 0.8%, *p* < 0.01, Figure 4). Preservation of myocardial architecture in the treatment group supports a role for hCitH3-mAb in attenuating adverse remodeling.

## 4. Discussion

This study provides direct experimental evidence that NETosis, through the release of CitH3, plays a pivotal role in the pathogenesis of acute MI and its progression to heart failure. By targeting CitH3 with hCitH3-mAb, we demonstrate a robust therapeutic effect –attenuating myocardial IRI, preservation of mitochondrial integrity, and sustained improvement in cardiac function post-MI. Mechanistically, CitH3 neutralization suppresses pro-inflammatory cytokine signaling (IL-1β and IFN-β), and preserves the peri-infarct microenvironment, thereby limiting infarct expansion and adverse remodeling. To the best of our knowledge, we provide the first evidence that hCitH3-mAb therapy can hinder the transition from MI to HF, underscoring its promise for future translational and clinical applications.

Neutrophils, the initial effectors of innate immunity, quickly migrate to areas of sterile tissue damage, including those caused by myocardial infarction [2,17]. While the formation of NETs is critical for host defense and clearance of necrotic debris, excessive NETosis generates damage-associated molecular patterns (DAMPs), including CitH3, which can exacerbate tissue injury by compromising macrophage function, damaging endothelial cells, and perpetuating sterile inflammation [34,35]. During acute MI, neutrophils are rapidly recruited to the infarct zone by cellular debris and DAMPs released from necrotic cardiomyocytes [36,37,38]. Our study shows that CitH3 levels rise sharply during early reperfusion and remain elevated for at least 14 days post-MI, implicating CitH3 as a key contributor to both acute injury and chronic remodeling. Notably, treatment with hCitH3-mAb markedly reduced circulating CitH3 levels, minimized infarct size, and sustained LVEF over the entire post-MI period.

In the acute phase of I/R injury, prophylactic administration of hCitH3-mAb resulted in a 36% reduction in infarct size compared to controls, accompanied by a significant decrease in neutrophil infiltration and myocardial CitH3 expression. Inflammatory mediators, including IL-1β and IFN-β, elevated in untreated MI mice, were significantly suppressed by hCitH3-mAb. Furthermore, the observed reduction in plasma citrate synthase, an indicator of mitochondrial injury, suggests that CitH3 may directly contribute to mitochondrial dysfunction during reperfusion. These findings indicate that CitH3 neutralization not only limits immune-mediated injury but also preserves mitochondrial function during a critical window of cardiomyocyte vulnerability.

Heart failure is characterized by structural remodeling, including cardiomyocyte death, fibrosis, and persistent low-grade inflammation [14,39]. Chronic oxidative stress-induced NETosis may sustain this inflammatory milieu, further impairing cardiac function. Elevated CitH3 levels in patients with chronic HF correlate with disease severity, systemic inflammation, and gut microbiota dysbiosis [40]. Our data demonstrate that hCitH3-mAb provides long-term protection following MI, with treated mice showing significantly improved LVEF at 7, 14, and 21 days post-MI. Whereas control mice exhibited progressive decline in cardiac function, hCitH3-mAb–treated animals showed functional recovery following an initial drop on day 1. This early divergence highlights the importance of timely CitH3 blockade in modulating post-MI remodeling.

Mechanistically, hCitH3 attenuates acute MI injury by neutralizing CitH3 and alleviating NETosis. Moreover, mitochondrial preservation in the hCitH3-mAb group was observed, as evidenced by reduced citrate synthase expression in infarct and peri-infarct myocardium. Since mitochondrial dysfunction is a key driver of cardiomyocyte death and adverse remodeling [41,42,43,44,45], this finding further supports the pathogenic role of CitH3 in cardiac injury. Beyond the acute phase, hCitH3-mAb attenuated chronic myocardial inflammation, as evidenced by decreased fibrosis and preserved ventricular architecture at day 21 post-MI. These findings implicate CitH3 not only in acute myocardial damage but also in sustaining the chronic inflammatory environment that drives HF progression.

While this study establishes CitH3 as a key mediator of myocardial injury, the precise mechanisms by which CitH3 activates inflammatory cascades and disrupts mitochondrial function require further investigation. In addition to its role in mitochondrial protection, we have recently completed a comprehensive study demonstrating the efficacy of hCitH3-mAb in preserving macrophage function in the context of NETosis-driven inflammation, thereby enhancing the body’s innate immune defense. Specifically, we provide compelling evidence that hCitH3-mAb effectively suppresses the CitH3-mediated self-amplifying cytokine storm, protecting tissues from injury without compromising the essential innate immune functions of NETs [30].

Further studies assessing additional mitochondrial markers, elucidating the mechanism underlying hCitH3-mAb’s mitochondria protective function associated with MI and MI-HF transition, will be critical. Whether CitH3 directly injures cardiomyocytes or acts primarily through paracrine immune activation remains to be elucidated. Inhibiting peptidylarginine deiminase (PAD) enzymes, which catalyze histone citrullination, represents an alternative therapeutic strategy worth exploring [13,27,46]. Both PAD2 and PAD4 are peptidylarginine deiminases responsible for the citrullination of histone H3 (CitH3). In the context of MI, PAD4-mediated CitH3 has been implicated in NET formation and inflammatory cell infiltration, key contributors to adverse myocardial remodeling. Notably, PAD4 knockout mice exhibit resistance to MI-induced injury [47,48,49]. Building on these findings, our study highlights the pathological role of CitH3 in cardiac remodeling and identifies both PAD2 and PAD4 as upstream mediators of this process. Importantly, our hCitH3-mAb exhibits superior efficacy compared to commercially available anti-CitH3 antibodies in neutralizing CitH3 generated by both PAD2 and PAD4, thereby mitigating NETosis-driven inflammation and protecting cardiomyocytes from injury [10,29,30].

Additionally, comprehensive GLP-compliant toxicology studies in rodents and non-human primates will be necessary to establish the safety profile of hCitH3-mAb for systemic administration. Finally, although the current study provides strong efficacy data in rodent models, large animal studies (e.g., in swine) will be critical to validate the translational potential of hCitH3-mAb for human MI and HF.

In our recent publication in Nature Communications [30], we provided a comprehensive safety evaluation of hCitH3-mAb in mice, rats, and nonhuman primates, demonstrating a no observed adverse effect level (NOAEL) exceeding 200 mg/kg. These studies support the favorable safety profile of hCitH3-mAb and its advancement toward FDA IND filing and future human clinical trials.

## 5. Conclusions

In conclusion, our findings identify CitH3 as a critical effector of both acute myocardial injury and chronic post-MI remodeling. Neutralization of CitH3 with hCitH3-mAb interrupts this pathological cascade, preserves mitochondrial integrity, reduces inflammatory injury, and prevents long-term cardiac dysfunction. These results support CitH3 as a promising therapeutic target and lay the foundation for advancing hCitH3-mAb toward clinical translation for the treatment of MI and prevention of heart failure.

## Figures and Tables

**Figure 1 biomolecules-15-01196-f001:**
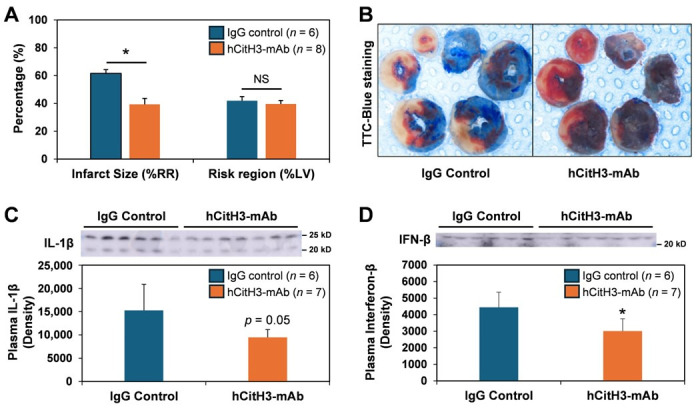
hCitH3-mAb attenuates myocardial infarct size associated with acute ischemia–reperfusion injury. (**A**), hCitH3-mAb attenuates myocardial infarct size following 40 min LCA occlusion and 24 h of reperfusion. (**B**), TTC-Blue staining to delineate infarcted, ischemic and non-ischemic regions. (**C**). Plasma level of IL-1β at 24 h of reperfusion. Western blot original images can be found in the Appendix A. (**D**), Plasma level of Interferon-β (IFN-β) at 24 h of reperfusion. NS, no significant difference. Western blot original images can be found in the Appendix A. * *p* < 0.05.

**Figure 2 biomolecules-15-01196-f002:**
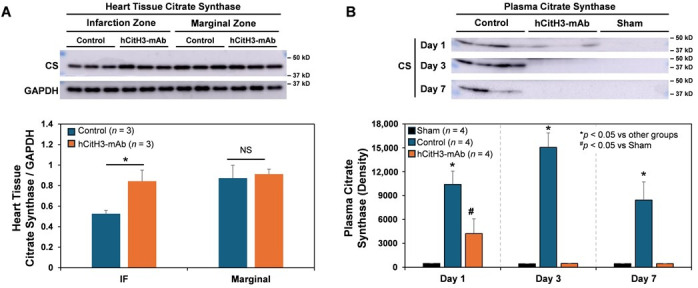
hCitH3-mAb treatment restores the expression of citrate synthase, a marker of mitochondrial integrity, in the injured heart. Western blot images and quantification of citrate Synthase (CS) levels in cardiac tissues (**A**) and plasma (**B**) in control and hCitH3-mAb-treated mice following 50 min of LCA occlusion and 72 h reperfusion. (**A**), myocardial CS levels were significantly decreased in the infarcted (IF) and peri-infarct marginal zones of IgG control mice, but preserved in hCitH3-mAb-treated mice. (**B**), plasma CS levels were markedly elevated on day 1, 3, 7 post-MI, but this increase was attenuated by hCitH3-mAb treatment. Thoracotomy-only sham mice showed minimal plasma CS. NS, no significant difference. Western blot original images can be found in the Appendix A. * *p* < 0.05.

**Figure 3 biomolecules-15-01196-f003:**
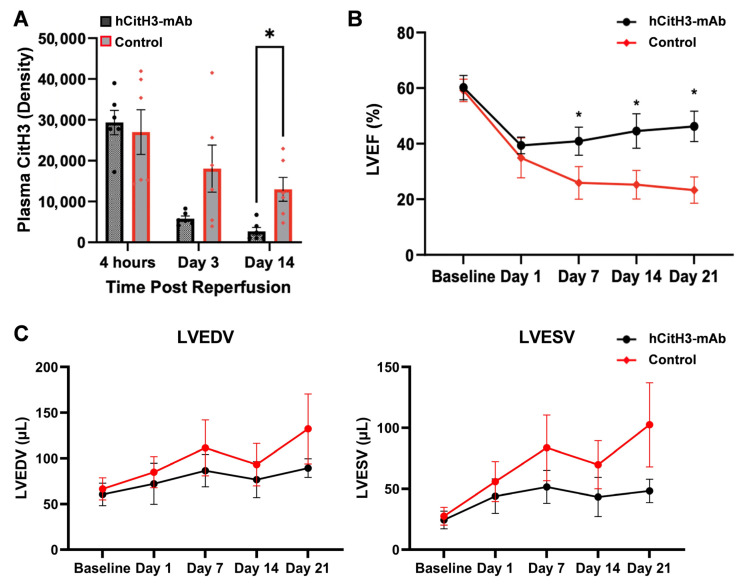
hCitH3-mAb neutralizes CitH3 to preserve cardiac function post-MI. (**A**), Quantification of Western blot data showed that plasma CitH3 levels were elevated at 4 h of reperfusion in both IgG control and hCitH3-mAb-treated mice and declined overtime, with a greater reduction in hCitH3-mAb-treated mice. The difference reached statistical significance on day 14. * *p* < 0.01. (**B**), left ventricular ejection fraction (LVEF), measured by ECHO, dropped in both groups on Day 1 but continued to decline in controls, while remaining significantly higher in treated mice throughout follow-up. * *p* < 0.01. (**C**), end-systolic LV volume (ESV) and end-diastolic LV volume (EDV) were elevated and continued to trend throughout the 4-week period of observation in both groups. The increase in LV volumes was higher in Control than in hCitH3-mAb-treated mice (*p* = NS).

**Figure 4 biomolecules-15-01196-f004:**
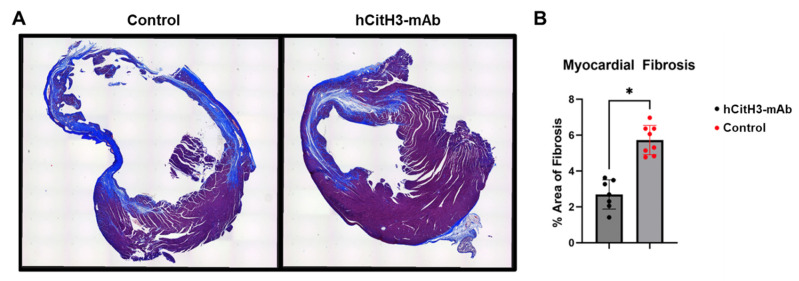
hCitH3-mAb mitigates fibrotic remodeling and preserves heart function post MI. (**A**), Masson’s trichrome staining of hearts at day 21 post-reperfusion showed less ventricular dilation and fibrosis in hCitH3-mAb–treated mice compared with controls. Collagen fibers appear blue, indicating fibrotic areas; muscle fibers are red; and cell nuclei are dark brown/black. (**B**), Quantification of fibrotic area (% cross-sectional area) showed a significant reduction in hCitH3-mAb-treated mice. * *p* < 0.05.

## Data Availability

The authors declare that the data supporting the findings of this study are available within the paper and its Appendix A.

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
