# Peer review of "Humanized Monoclonal Antibody Against Citrullinated Histone H3 Attenuates Myocardial Injury and Prevents Heart Failure in Rodent Models"

_biomolecules, 2025, doi:10.3390/biom15081196_

Round 1

Reviewer 1 Report

Comments and Suggestions for Authors

This study evaluated the effect of a humanized anti-CitH3 monoclonal antibody in a mouse model of myocardial infarction induced by left coronary artery oclusion followed by reperfusion to investigate the role of CitH3 in myocardial injury and adverse remodeling.

The study rationale is sound and the work has been nicely performed. 

However, the authors should address the following methodological and interpretation issues.

  1. The histological images of heart sections shown on Figure 4 are of low quality and do not represent clear differences in myocardial remodeling. The method used for fibrosis quantification should be described. The differences in ventricular dilation should be quantified or at least correlated with cardiac function parameters obtained by echocadiography.
  2. How do the authors explain the effect of the antibody on plasma levels  of CitH3? Does the antibody interfere with CitH3 detection by Western blot?
  3. The authors should discuss their findings in view of previous studies on PAD4 deficiency on myocardial remodeling. 

Author Response

Reviewer#1: 
This study evaluated the effect of a humanized anti-CitH3 monoclonal antibody in a mouse model of myocardial infarction induced by left coronary artery occlusion followed by reperfusion to investigate the role of CitH3 in myocardial injury and adverse remodeling. The study rationale is sound and the work has been nicely performed. However, the authors should address the following methodological and interpretation issues.

R1.1  The histological images of heart sections shown on Figure 4 are of low quality and do not represent clear differences in myocardial remodeling. The method used for fibrosis quantification should be described. The differences in ventricular dilation should be quantified or at least correlated with cardiac function parameters obtained by echocardiography.

Response: We appreciate the reviewer’s valuable feedback. In response, we have added a detailed description of the fibrosis quantification method to the Methods section (lines 161 - 163). We agree that the histological images in Figure 4 do not clearly reflect the extent of ventricular dilation observed via echocardiography. Due to tissue fixation in formalin, we are unable to accurately assess left ventricular volumes from histological sections, as fixation alters cardiac geometry and precludes distinguishing between systolic and diastolic phases.

To address this, we have now included additional echocardiographic data, specifically, end-diastolic volume (EDV) and end-systolic volume (ESV) in Figure 3C to better illustrate and quantify ventricular dilation, and to correlate functional changes with histological findings. Figure 3C was added to Figure 3. Description of the results was added to lines 222 – 225 and lines 233 – 235.

R1.2  How do the authors explain the effect of the antibody on plasma levels of CitH3? Does the antibody interfere with CitH3 detection by Western blot?

Response: We thank the reviewer for this important question. In our study, we did not observe a significant reduction in plasma CitH3 levels within the first 4 hours following reperfusion, suggesting ongoing NETosis and active CitH3 production in the myocardium during the acute phase. Treatment with hCitH3-mAb appears to disrupt the self-amplifying cycle in which extracellular CitH3 promotes further NETosis, thereby leading to a reduction in CitH3 levels during the later phases of reperfusion. To clarify, the observed lower CitH3 levels in plasma reflect the neutralization of free CitH3 by hCitH3-mAb, rather than a reduction in total CitH3 or antibody-mediated-masking. Once bound by hCitH3-mAb, CitH3 is functionally inactivated and no longer available to interact with other cellular targets or trigger downstream effects such as NETosis or pyroptosis. The key takeaway is that CitH3 bound to the hCitH3-mAb does not possess biological activity.

R1.3  The authors should discuss their findings in view of previous studies on PAD4 deficiency on myocardial remodeling.

Response: We thank the reviewer for this insightful comment. Both PAD2 and PAD4 are peptidylarginine deiminases responsible for the citrullination of histone H3 (CitH3). In the setting of myocardial infarction (MI), PAD4-mediated CitH3 has been implicated in NET formation and inflammatory cell infiltration, all of which contribute to adverse myocardial remodeling, as mice with knockout of PAD4 show resistance to MI (PMID: 40252995). In response to the reviewer’s suggestion, we have revised the Discussion section to include previous work on PAD4 deficiency and its impact on myocardial remodeling. We highlight the unique feature of our hCitH3-mAb in sequestering CitH3 generated by both PAD2 and PAD4 in combatting the NETosis-mediated inflammation and injury to the cardiomyocytes.

“Both PAD2 and PAD4 are peptidylarginine deiminases responsible for the citrullination of histone H3 (CitH3). In the context of MI, PAD4-mediated CitH3 has been implicated in NET formation and inflammatory cell infiltration, key contributors to adverse myocardial remodeling. Notably, PAD4 knockout mice exhibit resistance to MI-induced in-jury [47-49]. Building on these findings, our study highlights the pathological role of CitH3 in cardiac remodeling and identifies both PAD2 and PAD4 as upstream mediators of this process. Importantly, our hCitH3-mAb exhibits superior efficacy compared to commercially available anti-CitH3 antibodies in neutralizing CitH3 generated by both PAD2 and PAD4, thereby mitigating NETosis-driven inflammation and protecting cardiomyocytes from injury[10,29,30]”. This was added to lines 313 - 322.

Reviewer 2 Report

Comments and Suggestions for Authors

The manuscript presents compelling preclinical data on the role of citrullinated histone H3 (CitH3) in myocardial infarction and heart failure, yet several important limitations should be noted:

  1. The study is restricted to murine models without confirmation in larger animal models or early human translational data, limiting clinical applicability.
  2. While hCitH3-mAb shows mitochondrial protection and anti-inflammatory effects, the precise mechanisms linking CitH3 to mitochondrial injury and adverse remodeling are not fully elucidated, and alternative pathways were not examined.
  3. Some experimental groups (n = 6–10) are relatively small, which reduces statistical power and may increase the risk of type II errors.
  4. The 21-day post-MI follow-up may not sufficiently capture the long-term progression of post-infarction heart failure.
  5. No evaluation of potential adverse effects, immunogenicity, or off-target actions of hCitH3-mAb was performed, which is critical for translational potential.

Author Response

Reviewer 2:

The manuscript presents compelling preclinical data on the role of citrullinated histone H3 (CitH3) in myocardial infarction and heart failure, yet several important limitations should be noted:

R2.1  The study is restricted to murine models without confirmation in larger animal models or early human translational data, limiting clinical applicability.

Response: We appreciate the reviewer’s comment and fully recognize the importance of validating preclinical findings in larger animal models to enhance translational relevance. While the current study was conducted in murine models, it is translational in nature and was designed to establish a strong mechanistic foundation. Building on these results, we have developed a porcine model of myocardial infarction and have successfully established the technology to produce large quantities of hCitH3-mAb suitable for use in large animal studies. These efforts position us well to pursue the next phase of translational research aimed at bridging preclinical findings with potential clinical application.

R2.2  While hCitH3-mAb shows mitochondrial protection and anti-inflammatory effects, the precise mechanisms linking CitH3 to mitochondrial injury and adverse remodeling are not fully elucidated, and alternative pathways were not examined.

Response: We acknowledge that the precise mechanisms by which hCitH3-mAb confers mitochondrial protection remain to be fully elucidated as we already stated in the Discussion (line 308 - 311).

“In addition to its role in mitochondrial protection, we have recently completed a com-prehensive study demonstrating the efficacy of hCitH3-mAb in preserving macrophage function in the context of NETosis-driven inflammation, thereby enhancing the body’s innate immune defense. Specifically, we provide compelling evidence that hCitH3-mAb effectively suppresses the CitH3-mediated self-amplifying cytokine storm, protecting tissues from injury without compromising the essential innate immune functions of NETs[30].”

The above sentences have been added to the revised Discussion (see lines 301 - 307). We appreciate the reviewer for this important suggestion.

R2.3 Some experimental groups (n = 6–10) are relatively small, which reduces statistical power and may increase the risk of type II errors.

Response: While we acknowledge that larger sample sizes can increase statistical power and further reduce the risk of type II errors, we would like to point out that our current sample sizes are within the range commonly accepted in similar studies in the field. Importantly, our key findings have reached statistical significance, suggesting that the observed effects are robust despite the modest group sizes. We agree that increasing the sample size could further strengthen the study and will consider this in future experiments to reinforce the reproducibility and confidence in our findings.

R2.4  The 21-day post-MI follow-up may not sufficiently capture the long-term progression of post-infarction heart failure.

Response: We agree that long-term follow-up is crucial for fully characterizing the progression of post-infarction heart failure. In the current study, we chose the 21-day post-MI time point based on prior literature and our pilot studies, which showed that this period is sufficient to capture the early to intermediate phases of cardiac remodeling and functional decline. This timeframe also allowed us to assess the efficacy of our intervention during the critical window of early post-MI recovery. Nonetheless, we fully recognize the value of extended follow-up and plan to incorporate longer-term assessments in future studies to better understand the chronic progression of heart failure and the durability of therapeutic effects.

R2.5  No evaluation of potential adverse effects, immunogenicity, or off-target actions of hCitH3-mAb was performed, which is critical for translational potential.

Response: Thank you for raising these important suggestions for our pre-clinical studies with hCitH3-mAb.  In our recent publication in Nature Communications (Ouyang et al., 2025, in press), we provided a comprehensive safety evaluation of hCitH3-mAb in mice, rats, and nonhuman primates, demonstrating a no observed adverse effect level (NOAEL) exceeding 200 mg/kg. These studies support the favorable safety profile of hCitH3-mAb and its advancement toward FDA IND filing and future human clinical trials.

These statements have been added to the revised Discussion (lines 328 - 332).

Round 2

Reviewer 1 Report

Comments and Suggestions for Authors

Adequate answers to my comments

Reviewer 2 Report

Comments and Suggestions for Authors

None